# Development of a Scale of Nurses’ Competency in Anticipatory Grief Counseling for Caregivers of Patients with Terminal Cancer

**DOI:** 10.3390/healthcare11020264

**Published:** 2023-01-14

**Authors:** Chia-Chi Hsiao, Suh-Ing Hsieh, Chen-Yi Kao, Tsui-Ping Chu

**Affiliations:** 1Department of Nursing, Chiayi Chang Gung Memorial Hospital, Chiayi County 61363, Taiwan; 2College of Nursing, Taipei Medical University, Taipei City 11031, Taiwan; 3Department of Nursing, Chang Gung University of Science and Technology Chiayi Campus, Chiayi County 61363, Taiwan; 4Department of Nursing, Chang Gung University of Science and Technology, Taoyuan City 33303, Taiwan; 5Department of Nursing, Taoyuan Chang Gung Memorial Hospital, Taoyuan City 33378, Taiwan; 6Hospice and Palliative Care Ward, Taoyuan Chang Gung Memorial Hospital, Taoyuan City 33353, Taiwan

**Keywords:** patients with terminal cancer, caregivers, competency in anticipatory grief counseling

## Abstract

Anticipatory grief leads to a highly stressful and conflicting experience among caregivers of patients with terminal cancer. Nurses lack the competency to assess and manage the caregivers’ psychological problems, which in turn affects the caregivers’ quality of life. A scale assessing the anticipatory grief counseling competency among nurses is unavailable. In this study, an Anticipatory Grief Counseling Competency Scale (AGCCS) was developed for nurses. The Scale (AGCCS) was translated into Chinese and then revised. Psychometric testing of the scale was conducted on 252 nurses who participated in the care of patients with terminal cancer at a regional teaching hospital in Southern Taiwan. The data were analyzed using descriptive statistics, reliability, and Pearson’s correlation, and principal component analysis and analysis of variance were performed. Item- and scale-content validity indexes were 0.99 and 0.93, respectively. The Cronbach α of internal consistency was 0.981. The final 53-item AGCCS had five factors, which accounted for 70.81% of the total variance. The Pearson correlation coefficients of these factors ranged between 0.406 and 0.880 (*p* < 0.001). The AGCCS can be used to evaluate the aforementioned competency for improving caregivers’ quality of care. It can also facilitate in-service education planning and evaluation.

## 1. Introduction

In 2021, cancer was one of the 10 leading causes of death in Taiwan, accounting for 28.0% of all deaths [1]. An integrative literature review revealed that 12.5% to 38.5% of caregivers experience anticipatory grief symptoms before the death of their relatives [2]. In caregivers of patients with terminal cancer, anticipatory grief can result in a highly stressful and conflicting experience. Nurses lack the relevant competency for assessing and managing the psychological problems encountered by these caregivers. Consequently, they are unable to address the caregivers’ anticipatory grief promptly or appropriately, which in turn affects the caregivers’ quality of life [3]. Inadequate assessment and management of anticipatory grief in the caregivers of patients with terminal cancer may lead to complex grief reactions [4].

In clinical nursing education, the emphasis has shifted from obtaining information using a didactic method to enhancing problem-solving through experiential learning [5]. With competency-based education, nurses are taught to fulfill the demands of the clinical care environment; moreover, competency-based assessments are applied to evaluate their competency [6]. These assessments are performed using Miller’s pyramid of clinical competency and Kolb’s experiential learning cycle [6].

In Canada, hospice nurses are required to have core competencies, including assessing the needs of patients’ families and providing them with knowledge on loss, grief, and bereavement. Hospice nurses must consider the stage of development and aid patients’ families in recognizing the characteristics of grief and distinguishing grief and depression. The nurses must also identify family members at high risk of complex grief; assist them in predicting, recognizing, and adjusting their individual responses to loss and death; consider the unique needs of children at different developmental stages; and support the process of loss, grief, and bereavement through the grieving nursing care plan [7]. In Colombia, hospice nurses’ core competencies include communication; the formation of therapeutic interpersonal relationships with patients and families; and the provision of emotional, grief, and spiritual care to these individuals [8]. The core competencies outlined by the European Association for Palliative Care include the provision of patient- and family-centered care; comfort care; psychological, social, and spiritual care; the ability to cope with ethical challenges; inter-professional cooperation; communication skills; development of interpersonal relationships with patients and families; self-awareness; and commitment to continuous professional development [9]. In Taiwan, these competencies include spiritual care, life review, and death preparation [10]. Nurses play multiple roles in grief care in terms of patient- and family-centered care, advocacy, and professional development [11,12,13,14]. They are also responsible for assessing caregivers’ psychological state and providing them with emotional support by listening to them (to allow caregivers to express their emotions), and they also improve the sensitivity of grief assessment through grief counseling theory to support caregivers in managing anticipatory grief [15]. Among all healthcare providers, nurses have the most contact with patients and their families [16]. Therefore, they also have more opportunities to listen to patients (including those with terminal cancer) and caregivers and to assess their psychological distress levels. Nurses’ inability to provide such care can negatively affect their patients’ and caregivers’ quality of life [3].

Scales for measuring anticipatory grief counseling competency, however, have been developed only in the field of counseling. The Death Counseling Survey (DCS), designed by 34 grief counseling experts, is a self-assessment of grief counselors’ professional knowledge and treatment and assessment skills. It contains 58 questions scored on a 5-point Likert scale (ranging from 1 = noncompliance to 5 = full compliance). The Cronbach α of the five subscales ranges from 0.80 to 0.94. The Grief Counseling Competency Scale (GCCS) was developed by Cicchetti [17] and Cicchetti et al. [18], modified from the DCS and revised by 27 experts with >5 years of family grief counseling experience by using the Delphi method. The GCCS is a self-assessment questionnaire comprising 46 questions scored on a 5-point Likert scale (ranging from 1 = this does not describe me to 5 = this describes me very well). The questionnaire is divided into two sections comprising nine questions related to personal competency and grief and 37 questions related to skills and knowledge of grief counseling competency; the Cronbach α values of Section 1 and Section 2 are 0.79 and 0.97, respectively. Of the 37 questions in the second section, questions 9, 9, and 19 address conceptual skills and knowledge, assessment skills, and treatment skills, respectively; the Cronbach α values of the three subscales in Section 2 are 0.59, 0.60, and 0.60, respectively [17].

The GCCS developed by Cicchetti [17] is primarily intended for counselors with a master’s degree; it emphasizes the life experience, philosophy, and attitude toward death, diagnostic criteria of grief and its distinction from other diagnoses, use of specific death-associated words to discuss death-related matters, and ability to convey matters and explain death to children at various stages of death concept development. However, some GCCS items are unachievable for clinical nurses, and academic and in-service nurse education often lacks counseling courses. In addition, nurses are not expected to help caregivers understand the process and types of anticipatory grief, identify the manifestations of anticipatory grief, understand the differences between anticipatory grief and grief, or evaluate the risk of anticipatory grief. To the best of our knowledge, no scale has yet been developed for evaluating the competency of nurses in providing anticipatory grief counseling to caregivers of patients with terminal cancer. Therefore, in this study, we developed a scale for this purpose; psychometric testing of this scale was also conducted to evaluate the scale’s content and construct validity and internal consistency reliability.

## 2. Materials and Methods

### 2.1. Design

In this psychometric study, a cross-sectional survey was conducted using a structured questionnaire, which is part of a larger survey study [19].

### 2.2. Participants and Setting

Participants were recruited from the oncology ward of a regional teaching hospital in southern Taiwan by using convenience sampling [20]. A sample size of 3 to 6 per item was required. The participants included nurses, nurse practitioners, case managers, and assistant head nurses who had cared for patients with terminal cancer and were 20 years or older. After sampling was completed, the study instructions and anonymous questionnaires (with codes and commodity cards) were distributed among the participants. Of the 257 questionnaires issued, 251 completed questionnaires were returned, corresponding to a response rate of 97.7%.

### 2.3. Instruments

The instrument developed in this study featured sociodemographic and professional background characteristics and the Anticipatory Grief Counseling Competency Scale (AGCCS). The AGCCS was adapted from the GCCS [17] and employed care provided to caregivers in hospice care in Canada for their loss, grief, and bereavement, 7 spiritual care competencies [21], and spiritual care nursing interventions [22] as references for its development. The GCCS consists of 46 items scored on a 5-point Likert scale, and the scale is divided into 2 sections, with 9 items assessing perceived personal competencies (Cronbach α = 0.79) and 37 items assessing perceived skills and knowledge competencies (Cronbach α = 0.97). The 3 subsections of the second section include 9 items on conceptual skills and knowledge (Cronbach α = 0.59), 9 items on assessment skills (Cronbach α = 0.60), and 19 items on treatment skills (Cronbach α = 0.60) [17,18]. The GCCS has been adapted from the 58-item DCS, scored on a 5-point Likert scale; it contains 5 subscales: personal competencies (Cronbach α = 0.79), conceptual skills and knowledge (Cronbach α = 0.92), assessment skills (Cronbach α = 0.63), treatment skills (Cronbach α 0.87), and professional skills (Cronbach α = 0.83) [18].

Two experts with a nursing background and doctorate who had lived in English-speaking countries for >5 years performed the English-to-Chinese translation and Chinese-to-English back translation of the questionnaires used. The initial 60 items were evaluated for content validity by a panel of 7 experts with a master’s degree or higher; at least 5 years of experience in oncology, hospice, or palliative care; and who were ranked as an instructor or higher. These experts comprised 2 nursing supervisors, 1 nursing lecturer, 1 associate professor, 1 attending physician, 1 psychologist, and 1 social worker. Specifically, content validity was assessed as the topical appropriateness or relevance of the scale items on a 4-point Likert-type scale (ranging from 1 = not relevant to 4 = extremely relevant). The items were merged, added, separated, or revised according to the expert advice. After these revisions were made (55 items), a second round of expert validity review was performed, as an unfavorable scale-content validity index (S-CVI) of 0.68 was obtained. Further revision was conducted according to expert recommendations, as follows. “I can identify cultural differences in anticipatory grief care” and “I can identify cultural differences that affect responses to anticipatory grief” were combined into “I can identify cultural differences in anticipatory grief care and responses to anticipatory grief.” The final questionnaire comprised 55 items, including 9 items that addressed personal experiences of anticipatory grief, self-awareness, traits, skills related to counseling, and conceptual understanding of counseling. The remaining 46 items pertained to self-evaluation of knowledge and skills related to anticipatory grief counseling and nursing. All items were scored on a 5-point Likert scale (ranging from 1 = noncompliance to 5 = full compliance).

### 2.4. Procedure

Figure 1 illustrates the study procedure. After permission was obtained from the author of the GCCS, forward and backward translation of the 46-item GCCS was conducted. Then, the 51-item AGCCS was developed based on cited references and through the revision of items [5,17,18]. First and second rounds of expert content validity review were conducted.

A pilot study was conducted in the hematology, oncology, and neurosurgery departments (n = 29) between 26 November and 17 December 2019. Subsequently, the formal study was conducted in 11 units between 19 February and 6 March, 2020. Finally, the reliability and construct validity of the questionnaires were determined using the collected data.

### 2.5. Ethical Considerations

This study was approved by the study hospital’s institutional review board (201901235B0A3). Before the questionnaire was administered, the participants were assured that they could withdraw from the study at any time without consequences to their work assessment or promotion opportunities. They were also asked to provide informed consent. Except for the subject manual, the questionnaire content was replaced by codes to protect participant anonymity.

### 2.6. Statistical Analysis

All statistical analyses were performed using SPSS Statistics for Windows (version 21; IBM, Armonk, NY, USA). Assumptions of normality, linearity, outliers, and multicollinearity for factor analysis were checked [23]. For sociodemographic and professional characteristics and the AGCCS scores, descriptive statistics were analyzed for frequency, percent, mean, and standard deviation. Reliability analysis was used to assess the internal consistency of the AGCCS [24]. Principal component analysis (PCA) of the exploratory factor analysis with Promax rotation was used to examine construct validity, and these analyses were based on initial eigenvalues (≥1), factor loadings (>0.40), and scree plots [24,25]. Analysis of variance (ANOVA) was used to examine the mean difference in AGCCS scores on ever caring for family, relatives, and friends with anticipatory grief [26]. Pearson correlation was used to analyze the association between the AGCCS factors.

## 3. Results

### 3.1. Sociodemographic and Professional Characteristics of the Participants

Most of the participants were aged 26 to 30 years (31.1%), and nearly two-thirds were unmarried (65.3%). Moreover, 92% of the participants had a bachelor’s degree in nursing, and 68.5% had religious inclinations, including 39.5% and 17% with Taoist and folk beliefs, respectively. Most (71.3%) of the participants had experiences with death, and 66.5% of the participants had experienced anticipatory grief because of the death of a relative or through relatives and friends.

Most (71.3%) of the participants were clinical nurses. The participants worked in the hematology and oncology, pulmonology, gastrointestinal and hepatobiliary pathology, or urology departments. Moreover, 36.3% and 34.3% of the participants had 5 to 10 years of experience regarding working experience in the current department and the total working years. Of all the nurses, 42.6% had taken at least one course on or received training in anticipatory grief counseling during their education (26.7%) or as a part of the in-service programs at their workplace (16.7%).

### 3.2. Content Validity

The item-content validity index (I-CVI) and S-CVI of the AGCCS were 0.95 and 0.68 in the first round of expert content validity evaluation, respectively, and 0.99 and 0.93 in the second round of expert content validity evaluation, respectively (Figure 1).

### 3.3. Construct Validity—Principal Component Analysis of the Exploratory Factor Analysis

The 55 items on the scale were subjected to PCA with Promax rotation. The value of the Kaiser–Meyer–Olkin (KMO) measure of sampling adequacy was 0.965. The measure of sampling adequacy (MSA) of individual items ranged from 0.884 to 0.987. The Bartlett test of sphericity demonstrated significance (χ2 (1378) = 13,806.5, *p* < 0.001), with an initial eigenvalue and factor loading of >1.00 and 0.40, respectively. The items with corrected item-total correlations <0.30 (1–8, “I believe that there is more than one correct way to deal with anticipatory grief”) and factor loading <0.40 (1–9, “I have a sense of humor”) were excluded from the exploratory factor analysis step by step. Factors 1, 2, 3, 4, and 5 explained 52.18%, 8.08%, 5.40%, 2.67%, and 2.47% of the variance, respectively. In general, all five factors explained 70.81% of the total variance (Table 1). The scree plot in Figure 2 illustrates the five factors.

As presented in Appendix A (Table A1), the five factors of the scale were named as follows: factor 1 = competency in identification, assessment, and notification of anticipatory grief and enhancement of client’s self-expression and management (19 items); factor 2 = competency in nursing interventions for anticipatory grief (13 items); factor 3 = competency in counseling for anticipatory grief (nine items); factor 4 = personal experience, self-awareness, traits, and counseling perspective and competency in addressing anticipatory grief (seven items); and factor 5 = competency in respecting, accepting, and listening to anticipatory grief and inter-professional collaboration for anticipatory grief (five items). Most of the items loaded on factors 3 and 4 between the pattern matrix and structure matrix, except two items (AGCC2-19 and AGCC2-33), were identical. The item of AGCC2-19 was loaded on factor 1 of the pattern matrix and factor 5 of the structure matrix, whereas the item of AGCC2-33 was loaded on factor 1 of the pattern matrix and factor 2 of the structure matrix.

### 3.4. Construct Validity—Contrasted Groups Method

Our ANOVA results revealed that the mean AGCCS scores were significantly different between the participants who had (n = 138) and those who had not (n = 113) cared for family, relatives, or friends with anticipatory grief (176.47 ± 31.54 vs. 166.19 ± 20.09, F_(1, 249)_ = 7.07, *p* = 0.008).

### 3.5. Internal Consistency Reliability

In the pilot study (n = 29) and the formal study, the AGCCS had a Cronbach α of 0.975. and 0.981, respectively—demonstrating its internal consistency. The items with corrected item-total correlations <0.30 (1–8, “I believe that there is more than one correct way to deal with anticipatory grief”) and factor loading <0.40 (1–9, “I have a sense of humor”) were excluded during the exploratory factor analysis. As presented in Table 2, the internal consistency of the five factors ranged between 0.869 and 0.974. As shown in Table 2, two-tailed Pearson’s correlation revealed moderate to substantially high positive correlations between the factors (*r* = 0.406–0.880, *p* < 0.01). The correlation coefficients of factor 1 and factor 2 were the highest (*r* = 0.880, *p* < 0.01), followed by the correlation coefficients of factor 1 and factor 3 (*r* = 0.820, *p* < 0.01).

### 3.6. Descriptive Statistics of the AGCCS

Table 3 presents the grand means and means of the scores for the five factors of the AGCCS, as well as their corresponding standard deviations. The grand means and standard deviations for factors 1, 2, 3, 4, and 5 were 57.10 ± 13.21, 41.05 ± 8.79, 28.90 ± 6.19, 26.84 ± 4.00, and 17.95 ± 3.29 points, respectively. The means ± standard deviations of each factor divided by the number of items ranked from high to low were as follows: factor 4: 3.83 ± 0.57, followed by the means ± standard deviations of factor 5 (3.59 ± 0.66), factor 3 (3.21 ± 0.69), factor 2 (3.16 ± 0.68), and factor 1 (3.01 ± 0.70).

## 4. Discussion

In this study, we developed and evaluated the content and validity, and internal consistency of the AGCCS for nurses. As recommended by Polit and Beck [26], the evaluation by seven experts revealed that the scale had an I-CVI and S-CVI of 0.99 and 0.93, respectively—indicating high expert content validity of individual items and the overall scale, respectively.

Rather than principal axis factoring (PAF) and PCA with promax and direct oblimin, we used PCA with promax rotation, with an oblique solution for the rotation, because most item–item correlations were >0.30. [25] The KMO measure was 0.965, and the MSA of individual items ranged between 0.884 and 0.987 (i.e., >0.60) [25]. The Bartlett test of sphericity was significant (*p* < 0.001). These values indicate sampling adequacy and initial factor extraction. The 25 Factors 1, 2, 3, 4, and 5 explained 52.18%, 8.08%, 5.40%, 2.67%, and 2.47% of the variance, respectively; moreover, all five factors explained 70.81% of the total variance. Factor 1 explained 50% more variance than the other four factors. Although factors 3, 4, and 5 explained <5% of the variance, their eigenvalues were >1. In addition, the scree plot in Figure 2 presents all five factors [25]. We also reported the pattern and structure matrices for obliquely rotated solutions and named the five factors according to their items [25].

The results of the contrasted group’s method demonstrated that the mean AGCCS scores were significantly higher for the participants who had cared for family, relatives, or friends with anticipatory grief than those who had not (176.47 vs. 166.19). The contrasted group’s method was used to confirm the construct validity of the AGCCS, whereby the validity was examined on the basis of the degree to which the AGCCS demonstrated different scores for the groups known to have varying AGCCS scores [26]. Nurses who have cared for family, relatives, or friends with anticipatory grief may learn how to assess, plan, manage, and evaluate anticipatory grief by reflecting and working on improving their anticipatory grief counseling competency. Real-word experience presents powerful learning opportunities by situating nurses in authentic settings relevant to their clinical roles [5]. Competency in anticipatory grief counseling varies considerably depending on whether the nurses had cared for family, relatives, or friends with anticipatory grief, but not depending on their age. These results are identical to those of Cicchetti et al. [18] and Hsieh et al. [21]. In recent years, grief-related courses have been added to the curriculum of nursing schools in Taiwan. In the past, no similar courses or anticipatory grief courses were available, which resulted in insufficient nurse willingness toward and competency in anticipatory grief counseling. Presumably, because of their own experiences with anticipatory grief, nurses became highly aware of anticipatory grief reactions and their consequences. As a result of their empathy, they became more willing to offer anticipatory grief counseling for others with identical problems. Because of their anticipatory grief experiences or self-awareness and traits, they became more likely to focus on anticipatory grief responses in patients with terminal cancer and their caregivers. Therefore, they became more aware of anticipatory grief responses and their importance. These outcomes are similar to those reported in a previous study [27].

In the pilot and formal studies, the Cronbach α of the overall scale was 0.975 and 0.981, respectively. The internal consistency reliability of the five factors ranged from 0.869 to 0.974, indicating high reliability (>0.80), and the reliability value was slightly higher than that reported by Cicchetti18 for the GCCS, and no multicollinearity was observed among the reported issues [24,26]. The high internal consistency reliability of the AGCCS may be attributable to the inclusion of more items than those included by Cicchetti [18] and because the scale specifically measures the participants’ anticipatory grief counseling competency and the homogeneity among the participants in this respect [28]. This is consistent with the results of the GCCS [17,18], which was developed to measure master-level counselors involved in rehabilitation counseling. The scale was developed using the core competencies of palliative care in Canada [7] as well as spiritual care competency [21] and spiritual care nursing interventions [22] as references.

In this study, a substantially strong correlation [29] was observed between factors 1 and 2 and between factors 1 and 3. This can be explained by the fact that nurses are better able to provide anticipatory grief interventions and conduct anticipatory grief counseling when they can recognize, assess, and validate anticipatory grief and enhance clients’ self-expression and management skills in nursing care. Regarding the means ± standard deviations of each factor divided by the number of items, factor 4 showed the highest mean, followed by factors 5, 3, 2, and 1. Nurses usually possessed personal experience, self-awareness, traits, counseling perspective, and competency of anticipatory grief, as well as competency in respecting, accepting, and listening to anticipatory grief and inter-professional collaboration for anticipatory grief if their workload was low. Thus, they rated factor 4 and 5 items higher than factor 1 to 3 items. The mean of factor 1 was the lowest because the participants lacked training on the identification, assessment, and notification of anticipatory grief and enhancement of clients’ self-expression and management at school or in-service education. This competency can be strengthened through a series of workshops and bedside teaching methods (e.g., Objective Structured Clinical Examination and trusted professional activities based on competency) [30,31]. However, a single training session is insufficient for nurses to gain confidence in their grief counseling skills and conceptual knowledge [32].

This study has several limitations. First, this study was conducted in a single hospital, limiting the generalizability of the results. Nevertheless, the AGCCS may be used by other hospital’s nurses to assess nurses’ anticipatory grief counseling competency in the future and design in-service education programs. Second, the 53-item AGCCS was developed and tested in this study. Future studies can shorten it and test the construct validity of the shortened scale by using confirmatory factor analysis.

## 5. Conclusions

The five factors assessed for the AGCCS were significantly correlated and accounted for 70.81% of the total variance. The scale exhibited satisfactory content validity, internal consistency reliability, and construct validity. The AGCCS can be used to measure nurses’ competency in anticipatory grief counseling for patients with terminal cancer, their caregivers, or patients with acute trauma, thus improving their quality of care. Therefore, the AGCCS can aid in the planning and evaluation of in-service education.

## Figures and Tables

**Figure 1 healthcare-11-00264-f001:**
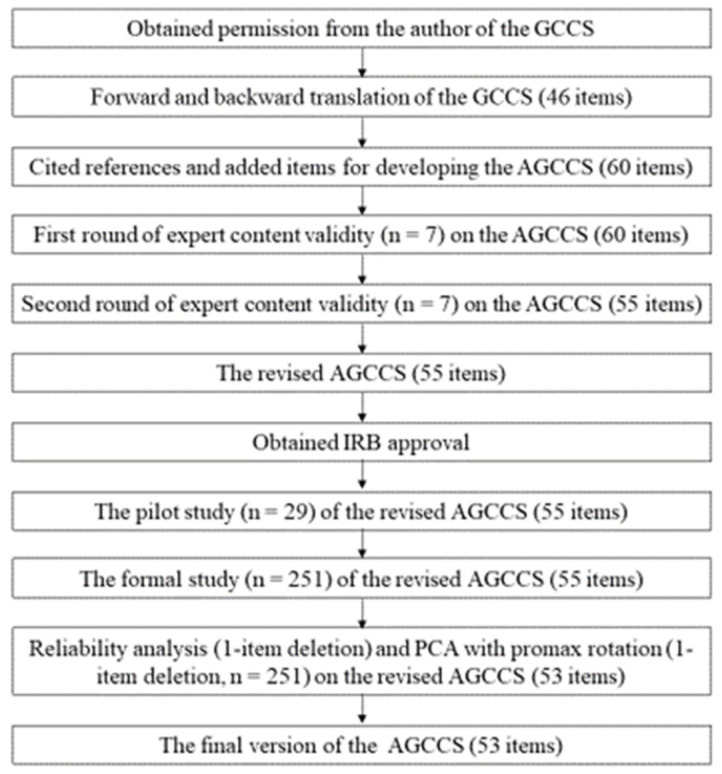
Study procedure.

**Figure 2 healthcare-11-00264-f002:**
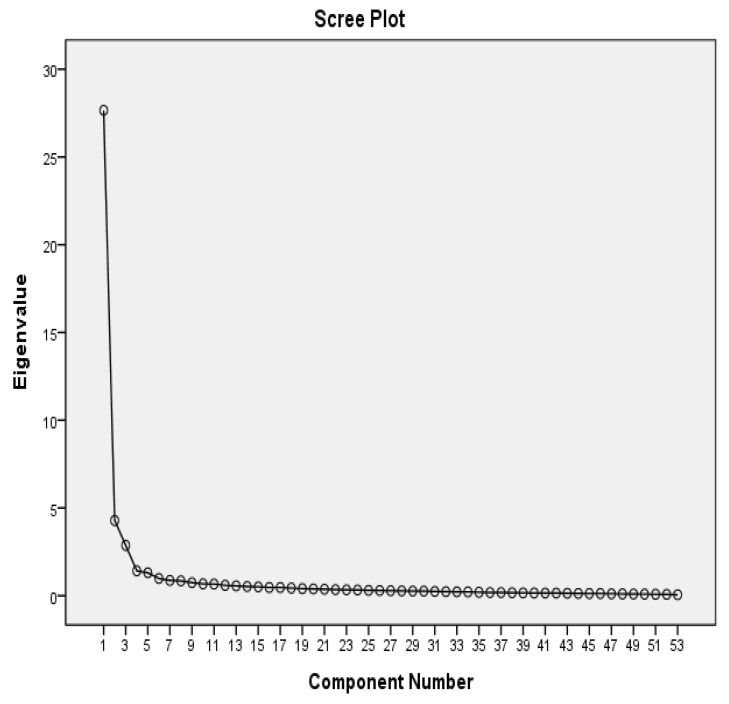
Scree plot.

**Table 1 healthcare-11-00264-t001:** Total variance explained by the five factors of the Anticipatory Grief Counseling Competency Scale (n = 251).

Factor	Initial Eigenvalues			Extraction Sums of Squared Loadings		
	Total	Variance (%)	Cumulative Variance (%)	Total	Variance (%)	Cumulative Variance (%)
Factor 1: Competency in identification, assessment, and notification of anticipatory grief and enhancement of client’s self-expression and management	27.66	52.18	52.18	27.66	52.18	52.18
Factor 2: Competency in nursing interventions of anticipatory grief	4.29	8.08	60.27	4.29	8.08	60.27
Factor 3: Competency in counseling of anticipatory grief	2.86	5.40	65.67	2.86	5.40	65.67
Factor 4: Personal experience, self-awareness, traits, and counseling perspective and competency in addressing anticipatory grief	1.42	2.67	68.34	1.42	2.67	68.34
Factor 5: Competency in respecting, accepting, listening to, and interprofessional collaboration for anticipatory grief	1.31	2.47	70.81	1.31	2.47	70.81

**Table 2 healthcare-11-00264-t002:** Pearson correlation coefficients between the five factors of the Anticipatory Grief Counseling Competency Scale (n = 251).

Factor	F1	F2	F3	F4	F5
Factor 1: Competency in identification, assessment, and notification of anticipatory grief and enhancement of client’s self-expression and management	1.000				
Factor 2: Competency in nursing interventions of anticipatory grief	0.880 **	1.000			
Factor 3: Competency in counseling of anticipatory grief	0.820 **	0.747 **	1.000		
Factor 4: Personal experience, self-awareness, traits, and counseling perspective and competency in addressing anticipatory grief	0.406 **	0.445 **	0.528 **	1.000	
Factor 5: Competency in respecting, accepting, listening to, and interprofessional collaboration for anticipatory grief	0.519 **	0.623 **	0.463 **	0.518 **	1.000

Note. ** *p* < 0.01 level (2-tailed).

**Table 3 healthcare-11-00264-t003:** Grand means, means, and standard deviations of the five factors of the Anticipatory Grief Counseling Competency Scale (n = 251).

Factor	Grand Mean (SD)	Item	Mean (SD)
Factor 1: Competency in identification, assessment, and notification of anticipatory grief and enhancement of client’s self-expression and management	57.10 (13.21)	19	3.01 (0.70)
Factor 2: Competency in nursing interventions of anticipatory grief	41.05 (8.79)	13	3.16 (0.68)
Factor 3: Competency in counseling of anticipatory grief	28.90 (6.19)	9	3.21 (0.69)
Factor 4: Personal experience, self-awareness, traits, and counseling perspective and competency in addressing anticipatory grief	26.84 (4.00)	7	3.83 (0.57)
Factor 5: Competency in respecting, accepting, listening to, and interprofessional collaboration for anticipatory grief	17.95 (3.29)	5	3.59 (0.66)

Abbreviation: SD, standard deviation.

## Data Availability

The datasets used and analyzed in this study are not available be-cause of the ethics restrictions set by the IRB.

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
