# Peer review of "Development of a Scale of Nurses’ Competency in Anticipatory Grief Counseling for Caregivers of Patients with Terminal Cancer"

_healthcare, 2023, doi:10.3390/healthcare11020264_

Round 1
Reviewer 1 Report
The study regards a nurses’ anticipatory grief counseling competency scale. The Scale was translated to Chinese and then revised. Psychometric testing of the scale was conducted in 252 nurses at a regional teaching hospital in Southern Taiwan. It was conducted in a single hospital, limiting the generalizability of the results, but they are very interesting. The most interesting result is that the mean AGCCS scores were significantly different between the participants who had and those who had not cared for family, relatives, or friends with anticipatory grief. They may learn how to assess, plan, manage, and evaluate anticipatory grief by reflecting and working on improving their anticipatory grief counseling competency. Real-word experience presents powerful learning opportunities by situating nurses in authentic settings relevant to their clinical roles. Based on these results, It could be to further explain the construct the Pearson correlation: in particular about values between factor 3 and 4, Competency in counseling of anticipatory grief and Personal experience, self-awareness, traits, as well as counseling perspective and competency of anticipatory grief. How the authors could explain these factors get .528 score if the experience is so important for the nurse’ competence? Another point is that participants were aged 26 to 30 years (31.1%): how the author could describe the relation between the experience and the competence with le curriculum of young participants?
Author Response
Point 1: The study regards a nurses’ anticipatory grief counseling competency scale. The Scale was translated to Chinese and then revised. Psychometric testing of the scale was conducted in 252 nurses at a regional teaching hospital in Southern Taiwan. It was conducted in a single hospital, limiting the generalizability of the results, but they are very interesting. The most interesting result is that the mean AGCCS scores were significantly different between the participants who had and those who had not cared for family, relatives, or friends with anticipatory grief. They may learn how to assess, plan, manage, and evaluate anticipatory grief by reflecting and working on improving their anticipatory grief counseling competency. Real-word experience presents powerful learning opportunities by situating nurses in authentic settings relevant to their clinical roles.
Response 1: Thank you for your comments.
Point 2: Based on these results, it could be to further explain the construct the Pearson correlation: in particular about values between factor 3 and 4, Competency in counseling of anticipatory grief and Personal experience, self-awareness, traits, as well as counseling perspective and competency of anticipatory grief. How the authors could explain these factors get .528 score if the experience is so important for the nurse’ competence?
Response 2: Pearson’s correlation analysis revealed moderate to substantially high positive correlations between the factors (r = .406–.880, p < 0.01). The correlation coefficients of factor 1 and factor 2 were the highest (.880, p < .01), followed by the correlation coefficients of factor 1 and factor 3 (r = .820, p < 0.01) (p. 6, lines 252–255).
Because of their anticipatory grief experiences or self-awareness and traits, nurses became more likely to focus on anticipatory grief responses in patients with terminal cancer and their caregivers. Therefore, they became more aware of anticipatory grief responses and their importance. These outcomes are similar to those reported in a previous study [27] (p. 8, lines 324–327).
Point 3: Another point is that participants were aged 26 to 30 years (31.1%): how the author could describe the relation between the experience and the competence with le curriculum of young participants?
Response 3: Our ANOVA results revealed that the mean AGCCS scores did not considerably vary with age. Competence in anticipatory grief counseling considerably varies depending on whether the nurses had cared for family, relatives, or friends with anticipatory grief, but not depending on their age. These results are identical to those of Cicchetti et al. [18] and Hsieh et al. [21]. In recent years, grief-related courses have been added to the curriculum of nursing schools in Taiwan. In the past, no similar courses or anticipatory grief courses were available, which resulted in insufficient nurse willingness toward and competency in anticipatory grief counseling. Presumably because of their own experiences with anticipatory grief, nurses became highly aware of anticipatory grief reactions and their consequences. As a result of their empathy, they became more willing to offer anticipatory grief counseling for individuals with identical problems. Because of their anticipatory grief experiences or self-awareness and traits, they are more likely to focus on anticipatory grief responses in patients with terminal cancer and their caregivers. Therefore, they became more aware of anticipatory grief responses and their importance. These outcomes are similar to those reported in a previous study [27] (p. 8, lines 298–311).

Reviewer 2 Report
The abstract has been prepared according to scientific rules. The background adequately describes the phenomenon of the problem that is used as the research topic. the scale of the problem needs to be added to data related to existing phenomena. method according to research objectives. the results are quite well explained accompanied by statistical tests. the discussion needs to be added in-depth reviews accompanied by literature or other appropriate references, the conclusions have answered the research objectives

Author Response
Point 1: The abstract has been prepared according to scientific rules. The background adequately describes the phenomenon of the problem that is used as the research topic.
Response 1: Thank you for your comments.
Point 2: The scale of the problem needs to be added to data related to existing phenomena.
Response 2: The scale of the problem has been added to the Introduction (pp. 2–3, lines 94–105).
Point 3: Method according to research objectives. the results are quite well explained accompanied by statistical tests. 4
Response 3: Thank you for your comments.
Point 4: The discussion needs to be added in-depth reviews accompanied by literature or other appropriate references, the conclusions have answered the research objectives.
Response 4: The Discussion section has been revised (p.8, lines 298–311 and 324–328; p.9, lines 337–341).
